# Real-Time American Sign Language Interpretation Using Deep Learning and Keypoint Tracking

**DOI:** 10.3390/s25072138

**Published:** 2025-03-28

**Authors:** Bader Alsharif, Easa Alalwany, Ali Ibrahim, Imad Mahgoub, Mohammad Ilyas

**Affiliations:** 1Department of Electrical Engineering and Computer Science, Florida Atlantic University, 777 Glades Road, Boca Raton, FL 33431, USA; aibrahim2014@fau.edu (A.I.); mahgoubi@fau.edu (I.M.); ilyas@fau.edu (M.I.); 2Department of Computer Science and Engineering, College of Telecommunication and Information, Technical and Vocational Training Corporation, Riyadh 12464, Saudi Arabia; 3College of Computer Science and Engineering, Taibah University, Yanbu 46421, Saudi Arabia

**Keywords:** deep learning, YOLO11, MediaPipe, transfer learning, real-Time ASL recognition, human–computer interaction, AI for accessibility, assistive technology

## Abstract

Communication barriers pose significant challenges for the Deaf and Hard-of-Hearing (DHH) community, limiting their access to essential services, social interactions, and professional opportunities. To bridge this gap, assistive technologies leveraging artificial intelligence (AI) and deep learning have gained prominence. This study presents a real-time American Sign Language (ASL) interpretation system that integrates deep learning with keypoint tracking to enhance accessibility and foster inclusivity. By combining the YOLOv11 model for gesture recognition with MediaPipe for precise hand tracking, the system achieves high accuracy in identifying ASL alphabet letters in real time. The proposed approach addresses challenges such as gesture ambiguity, environmental variations, and computational efficiency. Additionally, this system enables users to spell out names and locations, further improving its practical applications. Experimental results demonstrate that the model attains a mean Average Precision (mAP@0.5) of 98.2%, with an inference speed optimized for real-world deployment. This research underscores the critical role of AI-driven assistive technologies in empowering the DHH community by enabling seamless communication and interaction.

## 1. Introduction

For millions of Deaf and Hard-of-Hearing individuals worldwide, communication barriers hinder their ability to engage effectively in everyday interactions. Traditional methods, such as sign language interpretation services, are often limited in availability, costly, and reliant on human interpreters. In an increasingly digital world, there is a growing need for intelligent assistive technologies that provide real-time, accurate, and accessible solutions to bridge the communication gap [1].

American Sign Language (ASL) is one of the most widely used sign languages, consisting of distinct hand gestures that represent letters, words, and phrases. However, existing ASL recognition systems often struggle with real-time performance, accuracy, and robustness across diverse environments. To address these limitations, this study introduces a real-time ASL interpretation system that integrates deep learning with keypoint tracking. By leveraging YOLOv11 for rapid hand gesture recognition and MediaPipe for precise landmark detection, the proposed approach enhances the efficiency and reliability of ASL interpretation [2,3].

At the core of our system, a built-in webcam functions as a non-contact optical sensor that captures real-time visual data. This vision-based sensor converts light into digital image frames, which serve as the primary input for gesture analysis. MediaPipe extracts 21 keypoints per hand from each frame to generate a skeletal representation, while YOLOv11 detects and classifies specific ASL alphabet letters based on these visual inputs. The webcam’s role as a sensing device enables the system to acquire gesture-based data continuously and reliably. It also ensures that the entire recognition pipeline—from sensing to classification—can operate in real time, even under different lighting conditions and backgrounds, using only standard hardware. This highlights the system’s practical viability as an accessible and scalable assistive technology.

In addition to enhancing communication accessibility, AI-driven ASL recognition systems have the potential to revolutionize various industries. In education, they can facilitate more effective learning environments for Deaf students, allowing them to interact seamlessly with teachers and peers. In healthcare, real-time ASL interpretation can bridge the gap between Deaf patients and medical professionals, ensuring better access to quality care [4]. Similarly, workplaces can become more inclusive by integrating ASL recognition tools, enabling Deaf employees to participate fully in discussions and decision-making processes. The integration of AI and deep learning in assistive technologies not only fosters inclusivity but also paves the way for more advancements in human–computer interaction, making communication more intuitive and accessible for individuals with hearing impairments [5,6].

The significance of this research lies in its potential to empower the Deaf community by providing an AI-driven tool that translates ASL gestures into text, facilitating smoother interactions in education, workplaces, healthcare, and social settings. By developing a robust and accessible ASL interpretation system, this study contributes to the advancement of assistive technologies aimed at promoting inclusivity and bridging communication barriers for the DHH population [7].

The key contributions of our research paper can be summarized as follows:Developed a real-time AI-driven ASL interpretation system capable of accurately spelling out names and locations through hand gesture recognition.Integrated YOLOv11 for gesture detection and MediaPipe for hand tracking, ensuring both high accuracy and fast inference speeds.Implemented keypoint-based annotation techniques to enhance the model’s ability to capture subtle variations in finger positioning, improving classification precision.Utilized a standard built-in webcam as a vision-based optical sensor for gesture data acquisition, demonstrating the system’s practical application in real-time human–computer interaction and AI-driven assistive technology.Achieved an impressive 98.2% mean Average Precision (mAP@0.5), demonstrating the system’s reliability in recognizing ASL letters.Curated and processed a large-scale dataset of 130,000 ASL hand gesture images, each annotated with 21 keypoints to enhance model learning and generalization.

The research paper is structured as follows: Section 2 and Section 3 discuss relevant research and background concepts, including Transfer Learning, YOLO, and MediaPipe, which are fundamental to real-time ASL alphabet recognition. Section 4 elaborates on the methodology used in this study, detailing dataset preparation, preprocessing, augmentation, annotation, and data balancing techniques. Section 5 presents and discusses the results obtained, evaluating model performance through precision–recall analysis, confidence-based evaluations, and confusion matrix analysis, along with the real-time system’s robustness and accuracy. Finally, a general conclusion is provided in Section 6.

## 2. Background

### 2.1. Transfer Learning

Transfer learning is a deep learning technique that adapts a model pre-trained on a large dataset to a new, related task with a smaller dataset. Instead of training from scratch, this approach leverages the pre-trained model’s knowledge, improving generalization and reducing training time. It is particularly advantageous in object detection, where pre-trained models learn fundamental visual features—such as edges, textures, and shapes—that can be fine-tuned for specific applications [8,9].

For this task, a pre-trained YOLO11 model, originally trained on the COCO dataset, serves as the base model. This model has already learned essential visual patterns, making it well suited for hand detection. By utilizing its embedded knowledge, transfer learning enables adaptation for ASL-specific detection. During this process, the backbone of YOLO11—responsible for extracting core visual features—remains frozen, preserving its learned representations. Since the lower convolutional layers are already optimized for detecting edges, textures, and object structures, retraining them is redundant. Freezing these layers retains essential knowledge, prevents unwarranted modifications, and reduces computational overhead, allowing the model to focus on refining task-specific features [10].

While the backbone remains unchanged, the Neck and Head layers are fine-tuned to specialize in detecting ASL alphabet signs. These layers play a critical role in refining feature representations, predicting bounding boxes, and accurately classifying hand gestures. Fine-tuning involves several optimization strategies, including using a lower learning rate for stable weight updates, restricting gradient updates to trainable layers to preserve pre-trained features, and applying data augmentation techniques (e.g., flipping, rotation, and lighting variations) to enhance generalization [11].

Throughout this phase, the model progressively learns to distinguish between ASL letters, refine bounding box predictions, and improve classification accuracy, ensuring the reliable recognition of hand gestures. See Figure 1 for a visual representation of the workflow of the transfer learning methodology using the YOLO model.

### 2.2. YOLO

YOLO (You Only Look Once) is a state-of-the-art real-time object detection model designed for high-speed and accurate detection. YOLOv11 is an advanced version that builds upon previous iterations with improved performance in localization, detection accuracy, and efficiency. It is particularly optimized for detecting small and complex objects, making it ideal for applications such as ASL hand gesture recognition [12].

#### 2.2.1. Detection Pipeline

YOLOv11 follows a single-stage detection pipeline, meaning it processes the entire image in one pass instead of multiple steps like region-based models (e.g., Faster R-CNN) [13].

The input image is divided into grid cells (e.g., a 640 × 640 resolution image is divided into 32×32 cells). Each cell is responsible for detecting objects within its region. YOLOv11 utilizes a deep convolutional neural network (CNN) to extract meaningful spatial features from the image, ensuring efficient object detection [14].

#### 2.2.2. Bounding Box Prediction

Each grid cell predicts multiple bounding boxes [15], including

(x, y): Center coordinates of the object.(w, h): Width and height relative to the image size.Confidence Score: Probability of an object being present.

Since objects may overlap, multiple bounding boxes are generated to handle these cases effectively.

#### 2.2.3. Class Prediction

Each bounding box is assigned a class probability score using softmax classification [16]. For ASL hand gesture detection, the class labels range from 0 to 25, corresponding to the A–Z alphabet.

#### 2.2.4. Non-Maximum Suppression (NMS)

YOLO follows a technique called Non-Maximum Suppression (NMS), where multiple boxes may predict the same object. NMS removes redundant detections by keeping only the highest-confidence detection while suppressing lower-confidence overlapping boxes [17].

#### 2.2.5. Final Output

The final output consists of localized bounding boxes and class labels. In ASL detection, YOLOv11 detects hands and assigns them to ASL letter categories based on keypoints, ensuring accurate real-time recognition.

Figure 2 illustrates the step-by-step process of YOLO, including grid cell division, bounding box regression, classification with confidence scores, and final object detection.

### 2.3. MediaPipe

MediaPipe is an open-source framework developed by Google for building cross-platform, high-performance machine learning (ML) pipelines for real-time applications. It provides pre-built solutions for various computer vision tasks, including hand tracking, face detection, object tracking, pose estimation, and gesture recognition [18].

The MediaPipe Hand Tracking system employs a two-stage ML pipeline for precise and efficient hand detection and tracking.

#### 2.3.1. BlazePalm Detector

BlazePalm is an optimized deep learning model for real-time palm detection. As a single-shot detector, it processes an entire image in one pass, enabling the fast and efficient localization of hands. It generates a robust bounding box around the detected hand, even in cases of partial occlusion, ensuring stable tracking for applications like gesture recognition and sign language interpretation [19,20]. Its speed and accuracy make it ideal for mobile and embedded systems.

#### 2.3.2. Hand Landmark Model

The Hand Landmark Model is a lightweight convolutional neural network (CNN) trained on diverse datasets to detect 21 keypoints across the hand, including the fingers and wrist. It outputs (x, y, z) coordinates for each keypoint, where z provides depth estimation. This enables 3D gesture analysis [21].

#### 2.3.3. Keypoint Representation

The 21 detected keypoints are stored in a normalized form [22] using the following representation:(1)$$K_{i}=\left(x_{i},y_{i}\right),\;\mathrm{for}\;i\in \left[0,20\right]$$
where

$K_{i}$ represents the *i*-th keypoint.$x_{i},y_{i}$ are the normalized coordinates within the range [0,1].

This normalization ensures consistency across different image resolutions and scales, making the keypoints adaptable for various machine learning applications [22,23].

Table 1 shows the keypoint IDs and the hand part associated with each ID.

Figure 3 illustrates the 21 keypoints tracked by MediaPipe’s hand tracking model, capturing the articulation of the fingers and wrist for gesture recognition and 3D hand analysis.

By integrating YOLOv11 with MediaPipe Hands, this study enhances real-time ASL alphabet recognition, improving both its accuracy and its computational efficiency.

## 3. Related Work

Research on real-time American Sign Language (ASL) interpretation has advanced significantly with deep learning, particularly through object detection frameworks such as YOLO. Recent studies demonstrate that high accuracy can be achieved for static hand gesture recognition (e.g., fingerspelling alphabets) in real time. However, challenges persist in continuous sign language recognition. Below, we summarize key studies, their methodologies, and performance metrics, followed by a comparison table.

Early deep learning approaches applied object detection models for classifying hand gestures. A study by [24] trained a YOLOv5 model on the MU Hand Images ASL dataset (2425 images) and achieved a recall of 97% and a precision of 95%, resulting in an F1-score of approximately 96%. The model also attained a mean Average Precision (mAP) of 98% at IoU 0.5 (mAP@0.5). However, this dataset featured only one signer, limiting the model’s generalizability and reducing its practical applicability in diverse user scenarios.

Expanding upon this, [25] utilized a larger custom dataset of 8000 images (covering 40 classes, including letters and numbers) to develop a YOLOv4-based recognizer. Their model achieved an mAP@0.5 of 98.0%, with a recall of approximately 96% and an F1-score of 96%. They reported real-time performance at 28.9 FPS. However, their evaluation was conducted using pre-recorded video rather than live webcam integration, limiting the assessment of its robustness in dynamic, real-world environments.

More recent advancements in YOLO variants have further improved recognition accuracy. The study by [26] introduced a hybrid approach that combined YOLOv6 for static signs and an LSTM for dynamic signs. Their YOLOv6 model, trained on a custom-collected dataset, achieved a test accuracy of about 95%, with a precision of 96% and a recall of 92% for ASL alphabet gestures (mAP 96%). This outperformed their sequential-sign model, where an LSTM applied to continuous sign sequences achieved an accuracy of 88–92%, underscoring that static gesture recognition is currently more advanced than continuous sign recognition.

The study by [27] explored the application of YOLOv9 in real-time ASL interpretation. Their study utilized a dataset of 1334 images for training, validation, and testing. Despite achieving high accuracy across precision and recall metrics, the limited dataset size presents challenges for building a reliable real-time interpretation system.

In a more recent study, ref. [28] further refined sign language detection with YOLOv8, focusing on the Indonesian Sign Language System (SIBI). Their study trained YOLOv8 on video and image datasets, employing extensive preprocessing and data mining techniques to improve model efficiency. The model, optimized for 168 layers with over 11 million parameters, achieved an average inference speed of 4.6 ms per image. Detection results demonstrated exceptionally high success rates, particularly for the letters D, F, N, O, and Q, which achieved up to 96% accuracy. Overall, 22 out of 26 letters displayed “excellent” detection results (above 90%), while four letters (H, M, T, Z) showed “good” detection results (86–89%). The study highlights the potential of YOLOv8 in enhancing real-time sign language recognition.

Despite these advancements, several research gaps remain. Many studies focus solely on isolated signs (individual letters or digits) using relatively uniform image datasets, often featuring a single signer in a controlled background. While achieving near-perfect accuracy in such settings is impressive, it does not necessarily translate to real-world applications involving multiple users, varying lighting conditions, and continuous signing. Performance significantly declines when extending to word-level or sentence-level ASL, which involves gesture transitions and motion. For instance, the LSTM-based continuous sign recognition model in [26] attained only 88% accuracy, highlighting the difficulty of fluid ASL recognition. Additionally, the lack of a standardized large-scale benchmark dataset for continuous ASL gestures impedes direct comparisons among studies, as most researchers curate their own datasets. Some models also require on-screen hands or specific camera setups, making it challenging to achieve comparable accuracy with arbitrary video feeds in diverse settings.

To address these limitations, we propose a novel YOLOv11-based real-time ASL recognition system integrated with MediaPipe keypoint tracking. This approach enhances precision and robustness, overcoming challenges faced in previous studies. Unlike prior works that relied on small or single-signer datasets, our model is trained on a large-scale dataset of 130,000 images, each annotated with 21 keypoints. This ensures improved generalization across diverse signers, hand shapes, and lighting conditions. Moving beyond static gesture recognition, our system facilitates real-time ASL translation, allowing users to interactively spell names, locations, and simple phrases with high accuracy. The proposed model achieves an mAP@0.5 of 98.2% with an inference speed of 1.3 ms per image, making it highly suitable for real-world applications. By integrating deep learning with keypoint tracking, our system advances AI-driven assistive technologies, bridging the gap between research and practical ASL interpretation tools. This work sets the stage for more robust, real-time sign language translation in diverse environments.

Table 2 below provides a comparative summary of representative studies, detailing real-time capability, dataset size, and key performance metrics. All listed works focus on the deep-learning-based recognition of ASL alphabet gestures and report performance on static gesture classification

## 4. Methodology

This study aims to develop a real-time American Sign Language (ASL) alphabet recognition system that accurately identifies and translates ASL hand gestures into text, enabling users to spell names and locations interactively. The proposed approach integrates YOLOv11 for gesture detection and MediaPipe for hand tracking to enhance precision and real-time responsiveness.

A major challenge in ASL recognition systems lies in distinguishing visually similar gestures, such as “A” and “T” or “M” and “N”, which often leads to misclassifications. Additionally, the dataset quality presents significant obstacles, including poor image resolution, motion blur, inconsistent lighting, and variations in hand sizes, skin tones, and backgrounds. These factors introduce bias and reduce the model’s ability to generalize across different users and environments [29].

To address these challenges, a large-scale dataset was curated, incorporating custom keypoint-based annotations to enhance the model’s ability to detect subtle variations in finger positioning. This ensures improved accuracy, robustness against variations in hand structure, and better adaptability to real-world conditions.

Figure 4 presents an overview of the data processing pipeline for the proposed real-time ASL recognition system. It illustrates the sequential stages, including data preprocessing, augmentation, annotation, and model training, leading to the final real-time recognition system.

### 4.1. Dataset Description

The ASL Alphabet Hand Gesture Dataset is a comprehensive collection of hand gesture images designed to train deep learning models for real-time ASL recognition. This dataset enhances computer vision models by enabling the precise classification of ASL alphabet gestures, thereby supporting real-time sign language interpretation systems.

Comprising 130,000 images, the dataset captures diverse hand gestures under varying conditions to improve model generalization. These conditions include

Different lighting environments: bright, dim, and shadowed.Various backgrounds: natural outdoor settings and indoor scenes.Multiple angles and hand orientations for robustness.

Each image is annotated with 21 keypoints, accurately marking critical hand structures such as fingertips, knuckles, and the wrist (see Figure 5). These annotations provide a skeletal representation of the hand, enabling the model to distinguish between similar gestures with greater precision.

For optimal training and evaluation, the dataset is systematically partitioned as follows:Training set: 80%.Validation set: 10%.Testing set: 10%.

This structured division ensures that the trained models achieve high performance in real-world ASL recognition applications, making the dataset a valuable resource for AI-driven sign language accessibility solutions.

### 4.2. Data Preprocessing

To standardize the dataset and enhance model performance, several preprocessing techniques were applied:Auto-orientation: Ensuring the correct alignment of images regardless of the original orientation.Resizing: Standardizing all images to 640 × 640 pixels for consistency.Contrast adjustment: Enhancing visual clarity using contrast stretching.Grayscale conversion: Applied to 10% of images to improve robustness to color variations.

### 4.3. Data Augmentation

To enhance generalization and mitigate overfitting, 30% of the dataset underwent augmentation with the following transformations:Flipping: Horizontal and vertical flips to introduce diverse orientations.Rotation: Random variations between −10° and +10° to simulate natural hand positioning.Brightness adjustment: Random modifications within ±15% to accommodate lighting variations.Grayscale conversion: Selectively applied to certain images to enhance the model’s focus on gesture structure.

These augmentations significantly improve the model’s ability to recognize ASL gestures across different environmental conditions, ensuring higher accuracy and reliability in real-time applications.

### 4.4. Data Annotation

To optimize YOLO for ASL hand gesture recognition, images were annotated using 21 keypoints extracted from hand landmarks. Unlike traditional bounding boxes that enclose the entire hand without precisely identifying its central structure, this approach prioritizes the 21 keypoints detected by MediaPipe. From these keypoints, a bounding box is computed based on their spatial distribution, ensuring it accurately encapsulates the gesture while preserving crucial articulation details. By leveraging this method, the annotation process enhances precision, as it focuses on the actual structure of the hand rather than just its outer boundaries, leading to more reliable recognition and classification.

The annotation process was automated and underwent the following steps:Hand Landmark Detection:
MediaPipe extracted 21 keypoints corresponding to joints and finger positions of the hand.The detected keypoints were scaled and normalized to maintain consistency across varying image resolutions.
Bounding Box Calculation:A bounding box was generated to enclose only the 21 keypoints that were previously captured on the hand by determining the minimum and maximum coordinates from the extracted keypoints.

#### 4.4.1. Bounding Box Computation

Since YOLO requires object detection labels in the format (x_center, y_center, width, height), the bounding box was calculated using the following mathematical expressions:

##### Finding the Bounding Box Extent

The bounding box must enclose all 21 detected keypoints. To determine its position, we compute the top-left corner $\left(x_{\min},y_{\min}\right)$ and bottom-right corner $\left(x_{\max},y_{\max}\right)$:$$x_{\min}=\min\left(x_{0},x_{1},\ldots ,x_{20}\right),\;y_{\min}=\min\left(y_{0},y_{1},\ldots ,y_{20}\right)$$$$x_{\max}=\max\left(x_{0},x_{1},\ldots ,x_{20}\right),\;y_{\max}=\max\left(y_{0},y_{1},\ldots ,y_{20}\right)$$

Here, $x_{\min}$ and $y_{\min}$ represent the smallest x and y values among all detected keypoints, defining the top-left corner of the bounding box. Similarly, $x_{\max}$ and $y_{\max}$ represent the largest x and y values, defining the bottom-right corner.

##### Computing Bounding Box Dimensions

The width and height of the bounding box are calculated as$$\mathrm{width}=x_{\max}-x_{\min},\;\mathrm{height}=y_{\max}-y_{\min}$$

These values represent the total span of the hand in the x and y directions.

##### Determining the Bounding Box Center

Since YOLO requires bounding box coordinates in a center-based format, the center of the bounding box is computed as$$x_{\mathrm{center}}=\frac{x_{\min}+x_{\max}}{2},\;y_{\mathrm{center}}=\frac{y_{\min}+y_{\max}}{2}$$

This ensures the bounding box is properly centered around the detected hand.

##### Normalization for YOLO Format

Given that keypoints are already normalized between 0 and 1 relative to image dimensions, the bounding box coordinates must also be constrained within the same range:$$x_{\mathrm{center}},y_{\mathrm{center}},\mathrm{width},\mathrm{height}\in \left[0,1\right]$$

This prevents values from exceeding the image boundaries, ensuring consistency across different image resolutions.

#### 4.4.2. Annotation Format for YOLO

Each detected hand was assigned a class label corresponding to the ASL letter and formatted according to YOLO’s labeling convention:Class Label: An integer (0–25) representing the ASL letter.21 Keypoint Coordinates: Normalized values stored as$$K_{i}=\left(x_{i},y_{i}\right),\;i\in \left[0,20\right]$$
where $K_{i}$ represents the *i*-th keypoint, and $x_{i},y_{i}$ are in the range [0, 1].Bounding Box: Stored in the YOLO format as$$\left(x_{\mathrm{center}},y_{\mathrm{center}},\mathrm{width},\mathrm{height}\right)$$

Figure 6 illustrates the effectiveness of the annotation process during batch processing. It shows bounding boxes that focus solely on the 21 keypoints, ensuring precise localization.

This structured annotation approach enables YOLO to detect and classify ASL gestures with greater precision, leading to a fine-tuned, real-time ASL recognition system optimized for robust gesture classification.

#### 4.4.3. Data Balancing

Balancing the dataset is essential for training YOLOv11 to detect and interpret the American Sign Language (ASL) alphabet in real time. Without balance, the model favors more frequent classes, leading to bias where certain letters are detected with high accuracy while others are misclassified or ignored. This results in unreliable performance, making the system unsuitable for users relying on ASL [30].

An imbalanced dataset also affects YOLOv11’s detection confidence. If certain letters appear more often, the model assigns higher confidence to them while lowering confidence for less frequent ones, leading to poor recall and inconsistent real-time detection. Balancing the dataset ensures uniform exposure, resulting in stable confidence scores and improved accuracy across all classes.

Overfitting is another critical issue. If the model encounters significantly more samples of some letters, it becomes biased toward those while failing to learn the characteristics of rare ones. This weakens its ability to recognize less frequent letters, reducing overall effectiveness [31].

To address these challenges, the upsampling technique has been utilized through data augmentation to balance the dataset to 5000 images per class. This ensures equal learning across all ASL letters, prevents bias, enhances generalization, avoids overfitting, and improves real-world performance. A well-balanced dataset is crucial for building a reliable, inclusive, and effective real-time ASL interpretation system.

Figure 7 illustrates the dataset’s distribution and the spatial characteristics of the bounding boxes, offering insight into the overall structure and organization of the data.

Top-left (Bar Chart): This plot confirms that the dataset is balanced, with each of the 26 ASL letters having an equal number of instances (5000 images per class). The uniform distribution ensures that no letter is overrepresented or underrepresented, preventing model bias.

Top-right (Bounding Box Overlays): This visualization displays multiple bounding boxes superimposed on each other. The tight clustering of bounding boxes suggests that the hand positions across different images are well aligned and consistently annotated, reducing variations that could impact the model’s learning.

Bottom-left (Heatmap of x-y Coordinates): This heatmap shows the spatial distribution of the hand’s center positions in normalized coordinates. The density around the center suggests that most hand gestures are captured within a specific region of the frame, making it easier for YOLOv11 to detect and recognize them accurately.

Bottom-right (Heatmap of Width–Height Distribution): This plot highlights the distribution of bounding box sizes. The concentration of values in a specific region indicates that the hands in the dataset mostly follow a uniform scale, ensuring that the model generalizes well without being biased toward specific sizes.

This visualization confirms that the dataset is structured optimally for YOLOv11 training, ensuring consistency across annotations and reducing variability in hand positioning and bounding box dimensions.

## 5. Results and Discussion

### 5.1. Model Performance and Evaluation Metrics

The performance evaluation of the proposed American Sign Language (ASL) alphabet recognition system was conducted using standard multiclass classification metrics, including precision, recall, F1-score, and mean Average Precision (mAP). The system is powered by the trained YOLO11 model, a deep learning architecture with 181 layers and 2,594,910 parameters, optimized for efficient real-time inference. With a computational complexity of 6.5 GFLOPs (Giga Floating Point Operations per Second) per inference, the model was rigorously tested on a balanced dataset of ASL alphabet hand gesture images. The training was conducted with 300 epochs, a batch size of 16, and an input image size of 640 × 640. The AdamW optimizer was employed to enhance training stability.

The chosen configuration ensures the model’s reliability, accuracy, and scalability for real-world ASL translation applications.

The model’s performance is summarized in Table 3.

These results demonstrate the model’s strong ability to accurately detect and classify ASL hand gestures with a high mean Average Precision (mAP@0.5) of 98.2%, confirming its effectiveness in a real-time sign language interpretation system.

### 5.2. Precision–Recall Analysis in Multiclass Classification

The precision–recall (PR) curve provides insight into how the model balances precision and recall across different confidence thresholds for a multiclass classification setting [32].

In a multiclass classification setting, the precision and recall for each class *c* are computed as(2)$${\mathrm{Precision}}_{c}=\frac{TP_{c}}{TP_{c}+FP_{c}}$$(3)$${\mathrm{Recall}}_{c}=\frac{TP_{c}}{TP_{c}+FN_{c}}$$
where

$TP_{c}$ (True Positives): Correctly classified instances of class *c*.$FP_{c}$ (False Positives): Instances incorrectly classified as class *c*.$FN_{c}$ (False Negatives): Instances of class *c* that were misclassified.

From Figure 8, we observe that precision remains above 0.98 across most recall values, confirming minimal false positive detections. The slight degradation in precision at extreme recall values suggests that the model maintains a conservative detection strategy.

### 5.3. Confidence-Based Evaluations for Multiclass Classification

To further analyze model performance, we examine the recall–confidence, precision–confidence, and F1–confidence curves [33].

#### 5.3.1. Recall–Confidence Curve

In a multiclass classification scenario, the recall–confidence curve illustrates how recall changes as the confidence threshold τ increases [34]. The global recall at a given confidence level τ is computed as(4)$$\mathrm{Recall}\left(\tau \right)=\frac{\sum_{c=1}^{N}TP_{c}\left(\tau \right)}{\sum_{c=1}^{N}TP_{c}\left(\tau \right)+FN_{c}\left(\tau \right)}$$

As seen in Figure 9, recall remains at its highest value (close to 0.99) for lower confidence thresholds, ensuring that most detections are retained. However, as the confidence threshold surpasses τ=0.8, recall starts to decline sharply. This behavior indicates a trade-off where the model prioritizes precision over recall, filtering out lower-confidence predictions to improve accuracy while reducing the number of retrieved positive instances. The sharp drop in recall near τ=0.99 suggests that only the highest-confidence predictions are retained, which can be useful for applications requiring high precision such ASL interpretation systems.

#### 5.3.2. Precision–Confidence Curve

The global precision at confidence τ is computed as(5)$$\mathrm{Precision}\left(\tau \right)=\frac{\sum_{c=1}^{N}TP_{c}\left(\tau \right)}{\sum_{c=1}^{N}TP_{c}\left(\tau \right)+FP_{c}\left(\tau \right)}$$

From Figure 10, precision remains above 0.90 across nearly all confidence levels. The model achieves perfect precision at confidence τ=0.985, indicating that high-confidence detections are nearly always correct.

#### 5.3.3. F1–Confidence Curve

The F1-score at confidence τ balances precision and recall [35]:(6)$$F1\left(\tau \right)=2\times \frac{\mathrm{Precision}(\tau )\times \mathrm{Recall}(\tau )}{\mathrm{Precision}(\tau )+\mathrm{Recall}(\tau )}$$

Figure 11 shows the optimal F1-score occurs at τ=0.99, suggesting the best trade-off between detection accuracy and reliability.

### 5.4. Confusion Matrix Analysis

The confusion matrix provides a class-wise breakdown of model performance [36].

For a multiclass, the confusion matrix element at (i,j) represents(7)$$CM_{ij}=P\left(y_{\mathrm{pred}}=j|y_{\mathrm{true}}=i\right)$$
where $y_{\mathrm{pred}}$ and $y_{\mathrm{true}}$ are the predicted and actual labels, respectively. Refer to Figure 12 for a visualization.

Key findings:The model demonstrates high classification accuracy with minimal misclassifications.Class-wise performance remains consistent, ensuring robustness across all ASL alphabet classes.Confusion is minimal, confirming the effectiveness of the YOLO11n architecture in distinguishing ASL gestures.

These findings confirm the model’s strong classification ability, making it a promising solution for real-time ASL translation applications.

### 5.5. Evaluation of a Real-Time ASL Interpretation System

Evaluating a real-time American Sign Language (ASL) interpretation system for users to spell names and locations requires a comprehensive assessment of key performance aspects, including accuracy, latency, robustness, and usability. To conduct this evaluation, we utilized a laptop’s built-in webcam for real-time hand tracking and ASL recognition.

In our system, the webcam serves as a non-contact vision-based sensor that captures gesture data in the form of image frames. These frames are processed using MediaPipe to extract 21 hand keypoints and passed to YOLOv11 for ASL gesture classification. This sensor-driven pipeline enables real-time processing under various lighting and background conditions using only standard hardware.

#### 5.5.1. System Specifications

The system was tested on a laptop with the following specifications:Operating Environment: Python 3.10.11 and PyTorch 2.2.1.Hardware: 12th Gen Intel Core i7-1260P CPU.

#### 5.5.2. Performance Analysis

Our analysis of image processing per frame revealed the following time distribution:Preprocessing: 1.1 ms.Inference (Model Prediction): 70.9 ms.Loss Calculation: 0.0 ms.Postprocessing: 0.5 ms.

#### 5.5.3. Robustness Across Different Conditions

To further evaluate the system’s effectiveness, we tested it under various conditions, including different hand shapes, skin tones, and background complexities. The system demonstrated strong generalization capabilities, accurately recognizing these variations with minimal error.

This robustness can be attributed to the integration of MediaPipe, which focuses solely on detecting 21 keypoints of the hand. Once the hand shape is determined, this information is passed to YOLOv11 for the final recognition process. Despite the two-step approach—first detecting the hand shape and keypoints and then processing the recognition—our model remains fast and highly efficient in detecting hand gestures and translating them into text.

#### 5.5.4. ASL Translation to Names and Locations

Figure 13 and Figure 14 illustrate the system’s ability to interpret ASL gestures into meaningful text.

These results highlight the system’s efficiency in real-time ASL interpretation, ensuring minimal latency while maintaining high accuracy across diverse conditions.

## 6. Conclusions

The development of assistive technologies for the Deaf and Hard-of-Hearing community is crucial in fostering inclusivity and eliminating communication barriers. This study presents a real-time ASL interpretation system that effectively integrates deep learning with keypoint tracking, demonstrating high accuracy and efficiency in recognizing ASL alphabet gestures. By leveraging YOLOv11 for gesture detection and MediaPipe for precise hand tracking, the proposed system ensures robust performance across diverse conditions.

Experimental results validate the system’s effectiveness, achieving a mean Average Precision (mAP@0.5) of 98.2% with minimal latency, making it highly suitable for real-time applications. The ability to translate ASL gestures into text in real time significantly enhances accessibility in various domains, including education, healthcare, and professional settings. Future work will focus on extending the system’s capabilities beyond individual letter recognition to full ASL sentence interpretation, enabling more natural and fluid communication. This will require advancements in sequence modeling techniques, such as recurrent neural networks (RNNs), transformers, or temporal convolutional networks (TCNs), to effectively capture contextual dependencies between consecutive gestures.

In conclusion, this research underscores the transformative potential of AI-driven assistive technologies in empowering the Deaf community. By bridging the communication gap through real-time ASL recognition, this system contributes to a more inclusive society where individuals with hearing impairments can interact seamlessly with the world around them, whether introducing themselves or indicating locations.

## Figures and Tables

**Figure 1 sensors-25-02138-f001:**
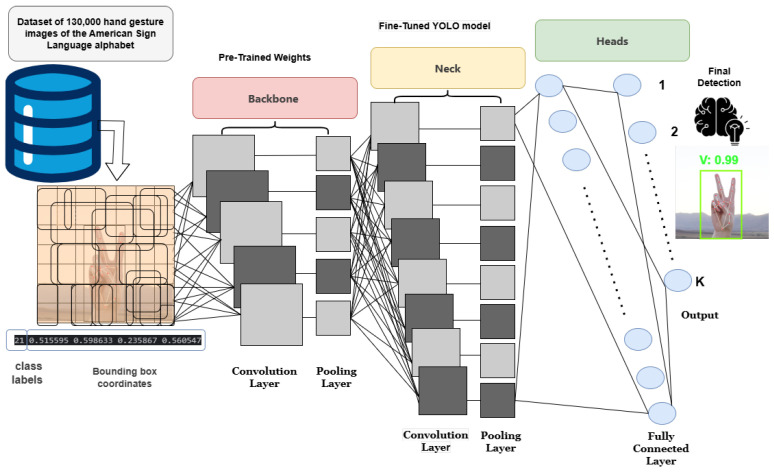
Workflow of transfer learning methodology for ASL hand gesture detection using YOLO.

**Figure 2 sensors-25-02138-f002:**
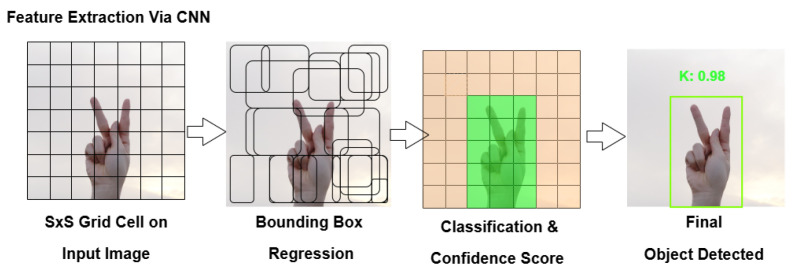
YOLO detection workflow for ASL hand gesture recognition.

**Figure 3 sensors-25-02138-f003:**
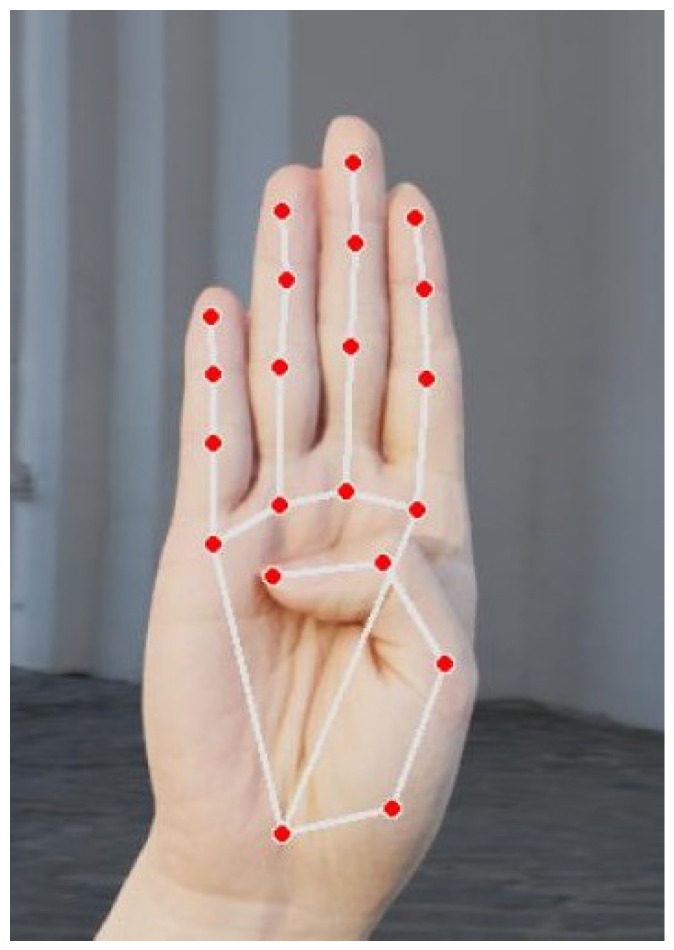
Illustration of the 21 keypoints tracked by MediaPipe’s hand tracking model.

**Figure 4 sensors-25-02138-f004:**
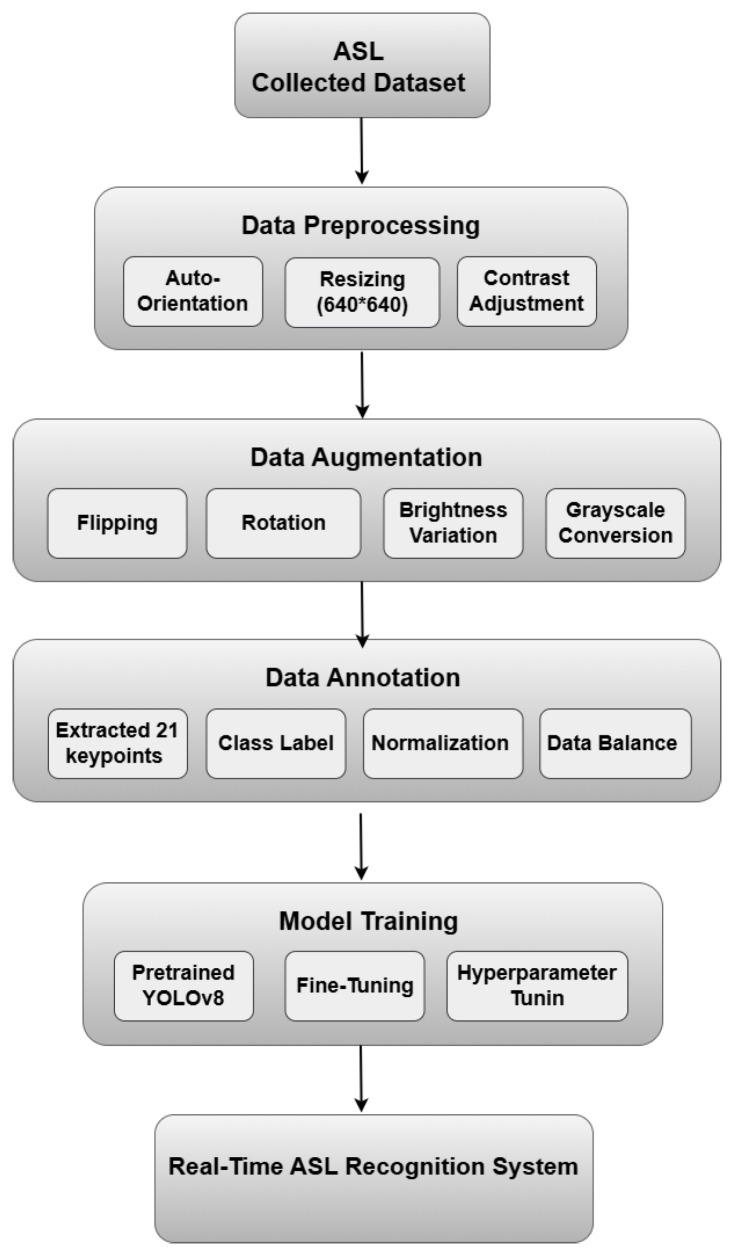
Workflow of the proposed ASL recognition model.

**Figure 5 sensors-25-02138-f005:**
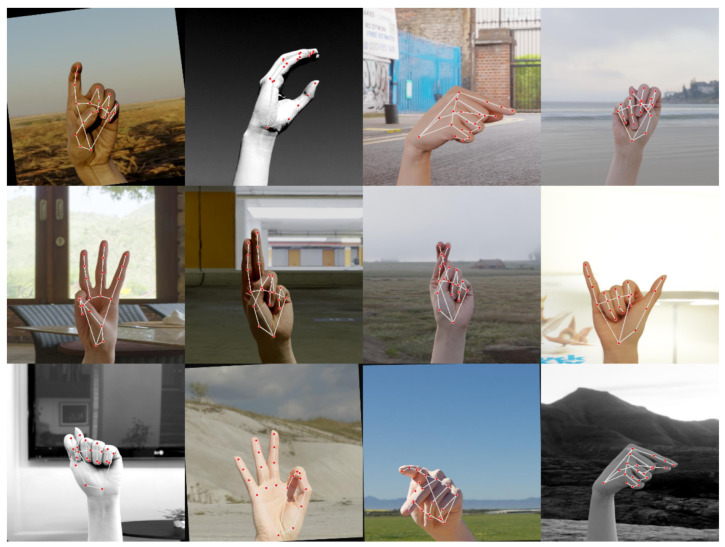
Annotated ASL Alphabet Hand Gestures with 21 keypoints for model training.

**Figure 6 sensors-25-02138-f006:**
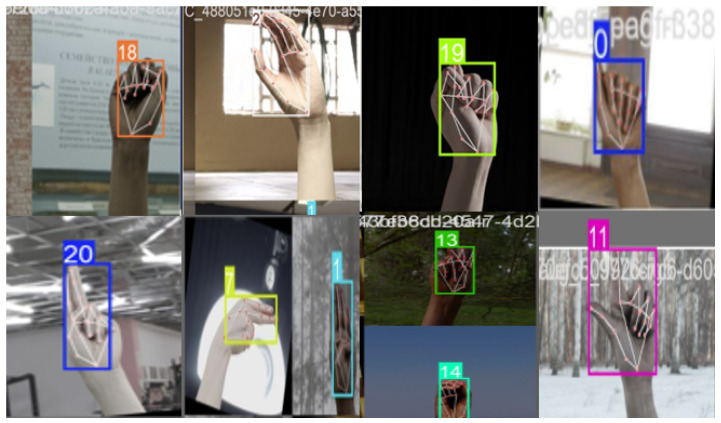
Effectiveness of annotation in batch processing with precise keypoint localization.

**Figure 7 sensors-25-02138-f007:**
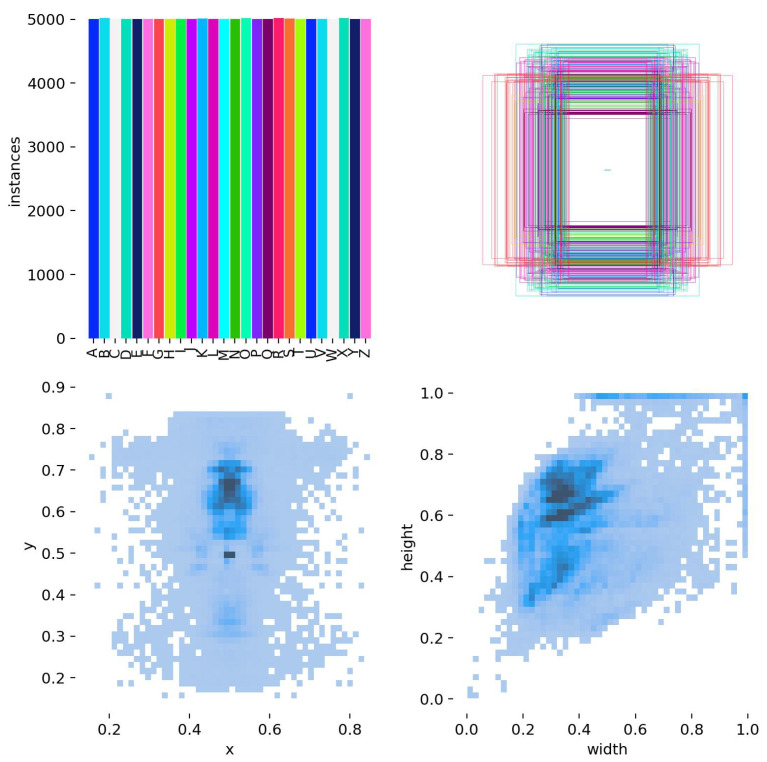
Overview of the dataset distribution and spatial properties of the bounding boxes.

**Figure 8 sensors-25-02138-f008:**
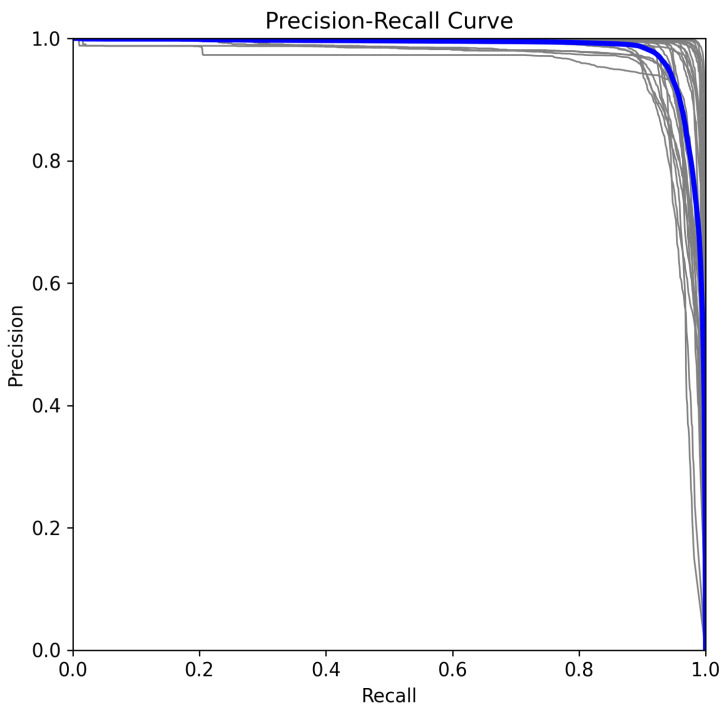
Precision–recall curve for ASL recognition model.

**Figure 9 sensors-25-02138-f009:**
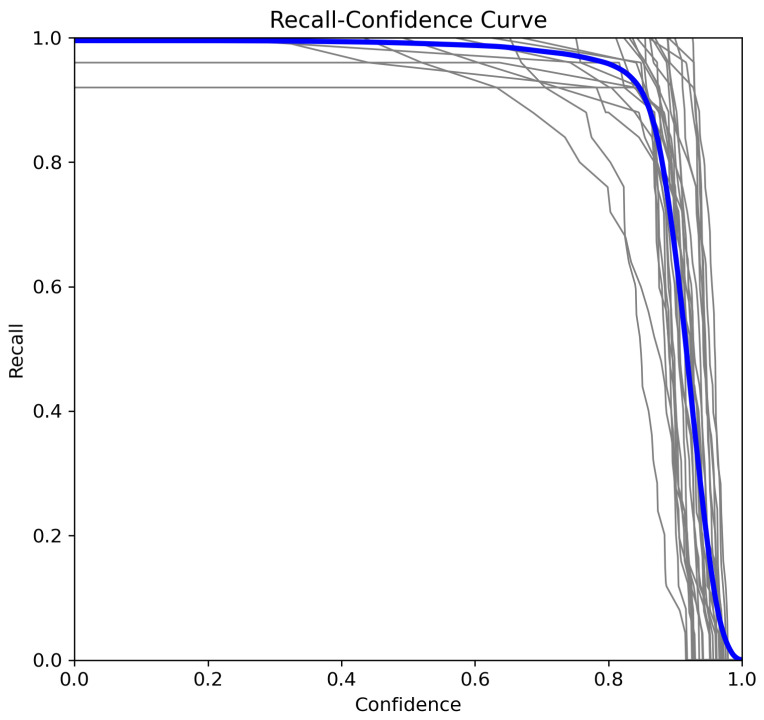
Recall–confidence curve.

**Figure 10 sensors-25-02138-f010:**
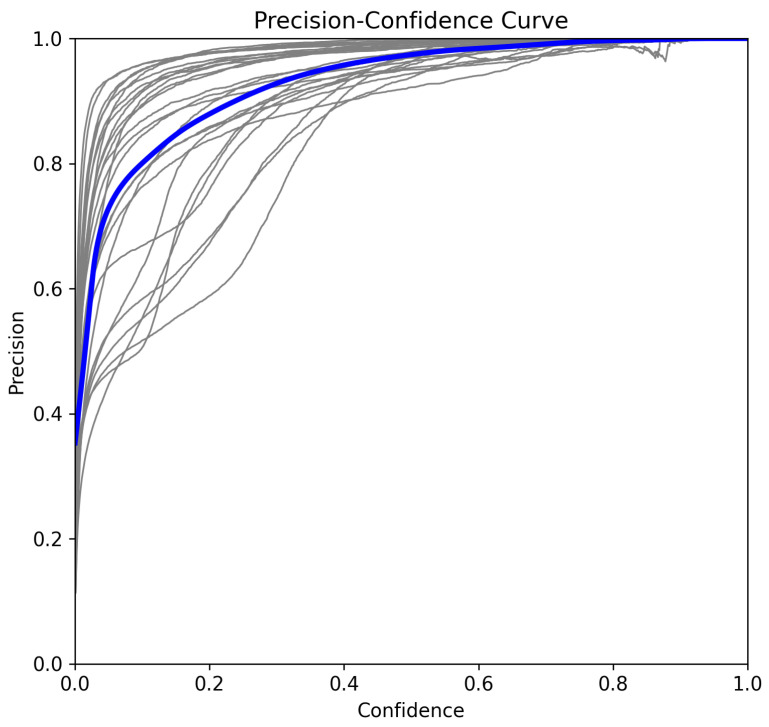
Precision–confidence curve.

**Figure 11 sensors-25-02138-f011:**
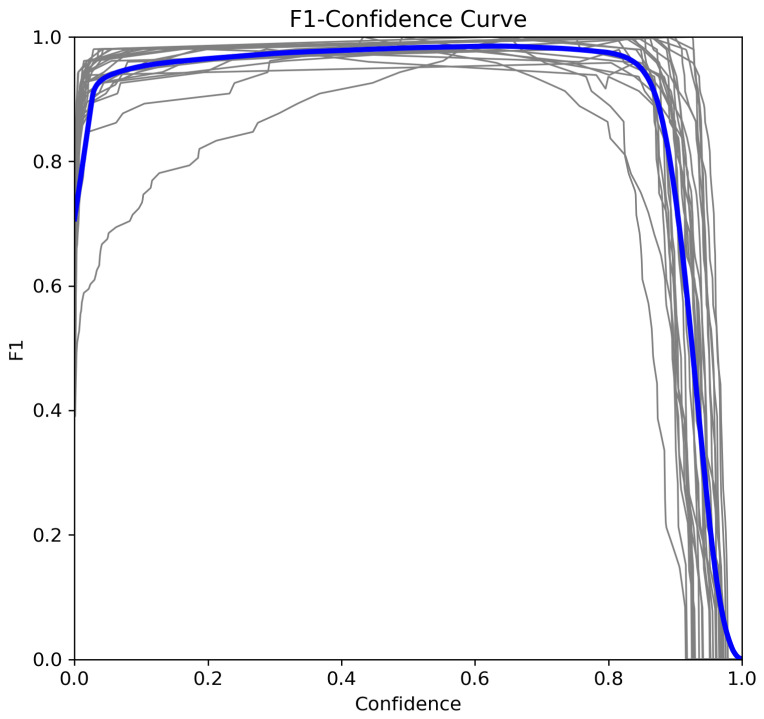
F1–confidence curve.

**Figure 12 sensors-25-02138-f012:**
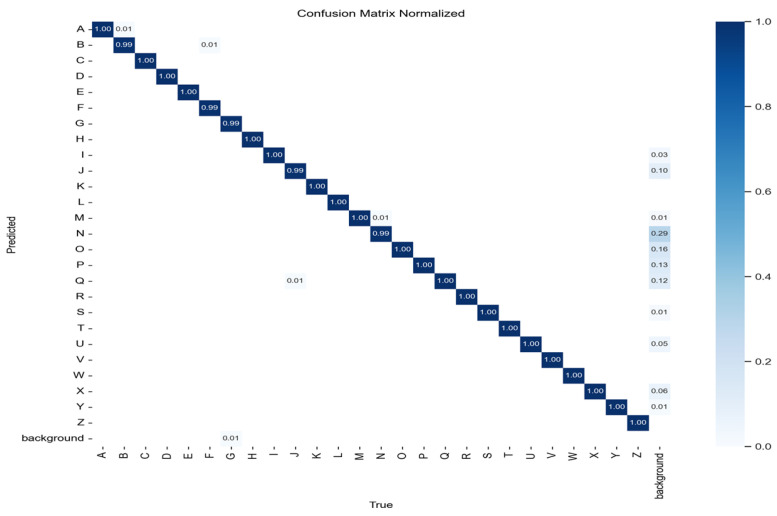
Normalized confusion matrix for ASL alphabet recognition.

**Figure 13 sensors-25-02138-f013:**
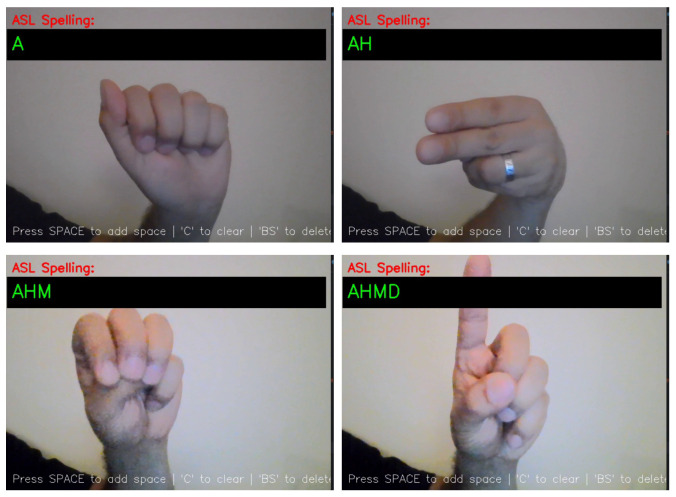
The model translating ASL into a name.

**Figure 14 sensors-25-02138-f014:**
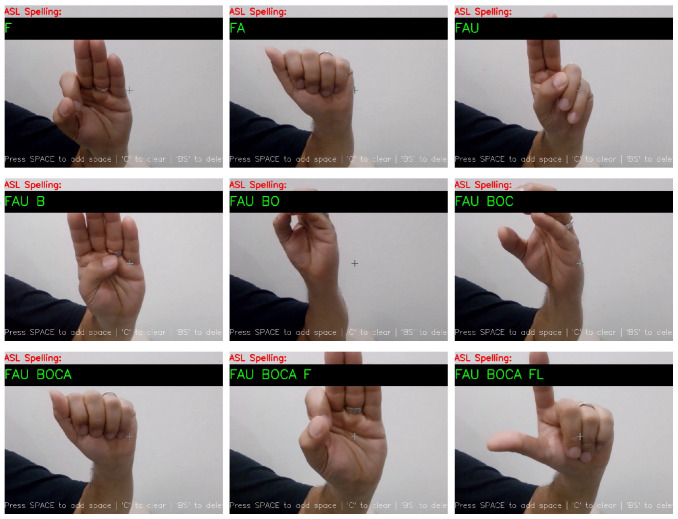
The model translating ASL into a location.

**Table 1 sensors-25-02138-t001:** MediaPipe hand keypoints and corresponding hand parts.

Keypoint ID(s)	Hand Part
0	Wrist
1, 2, 3, 4	Thumb
5, 6, 7, 8	Index Finger
9, 10, 11, 12	Middle Finger
13, 14, 15, 16	Ring Finger
17, 18, 19, 20	Pinky Finger

**Table 2 sensors-25-02138-t002:** Comparison of our proposed approach with recent advancements in YOLO-based ASL recognition models, highlighting dataset size, performance metrics, and real-time capability.

Ref	Year	Model	Dataset (Images)	Precision	Recall	F1-Score	mAP@0.5	Real-Time
[24]	2021	YOLOv5	2425	95%	97%	96%	98%	No
[25]	2022	YOLOv4	8000	96.00%	96.00%	96.00%	98.01%	Yes
[26]	2023	YOLOv6	8000	96.00%	96.00%	96.00%	96.22%	Yes
[27]	2024	YOLOv9	1334	96.83%	92.96%	-	97.84%	Yes
[28]	2025	YOLOv8	Video-based dataset	100%	99%	89%	96%	Yes
Our work	2025	YOLOv11	130,000	98.50%	98.10%	99.10%	98.20%	Yes

**Table 3 sensors-25-02138-t003:** Performance metrics of the ASL recognition model.

Metric	Value
Precision (P)	0.985
Recall (R)	0.981
F1-Sore (F)	0.991
mAP@0.5	0.982
mAP@0.5–0.95	0.933
Inference Speed	1.3 ms per image
Post-processing Speed	0.3 ms per image

## Data Availability

Data are available upon request.

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
