# Peer review of "Real-Time American Sign Language Interpretation Using Deep Learning and Keypoint Tracking"

_sensors, 2025, doi:10.3390/s25072138_

Round 1

Reviewer 1 Report

Comments and Suggestions for Authors

The article is devoted to the development of a real-time American language gesture recognition system that combines deep learning methods and tracking of key hand points. The authors use the YOLOv11 model for gesture recognition and MediaPipe to accurately track the positions of fingers and palms, which allows for more accurate segmentation of the hand in the frame and improve the accuracy of the advanced YOLO model. In addition, unlike existing studies, the authors use a newer version of YOLOv11, which also has a positive effect on gesture recognition over time. The study describes in sufficient detail the process of preparing the model, its further training, the operation of the models used, and data annotation. There are no comments on this section in this regard. The authors also indicate the specific tools used to solve the problem, the performance of the trained model, its evaluation by various metrics and in comparison with alternatives, which additionally confirms the effectiveness of the solution obtained. The list of sources is up-to-date, and an analysis of related studies has been conducted.
Thus, the presented work has high relevance and scientific significance. After reading it, I had one comment that is worth paying attention to:

In section 5.5.4, Figures 13 and 14 need to be improved. I understand that the authors wanted to demonstrate how the system works, but it's very difficult to understand from the images provided. My suggestion is to present these two figures as a sequence of 5-10 frames, from which you can see how the hand was recognized, what gesture was on the frame, and how the line with the recognized text gradually increases, replenished with new characters. 
In addition, it may be worthwhile to attach a screenshot in section 5.5.3 with the recognition of multiple gestures in poor light conditions.

Author Response

Thank you very much for your valuable feedback; it has been

instrumental in improving our research. Please see the attached.  

Reviewer 2 Report

Comments and Suggestions for Authors

The manuscript presents a real-time ASL interpretation system integrating YOLOv11 and MediaPipe, achieving high accuracy (98.2% mAP@0.5) and fast inference speeds. The work addresses critical challenges in assistive technologies for the Deaf and Hard-of-Hearing community, demonstrating significant potential for real-world applications in education, healthcare, and social interaction. While the study is well-structured and technically sound, several revisions are required before its publication.

1) The introduction and related work sections lack sufficient engagement with recent advancements in continuous sign language recognition (CSLR) and dynamic gesture interpretation. For instance, studies such as [Li et al., IEEE TPAMI 2023] on hybrid CNN-LSTM architectures for CSLR and [Wang et al., CVPR 2024] on transformer-based temporal modeling should be cited to highlight the novelty of the proposed static gesture recognition system and clarify its limitations compared to dynamic approaches.

2) The comparison table (Table 2) omits benchmark datasets like ASLLVD or WLASL, which are critical for evaluating generalization. 

3) Same to my first comment, the authors emphasize the novelty of this working using gesture recognition, the methods are interesting. In fact, there are lots of works related to the pattern recognition, including hardware and software tools. However, the authors seem not to realize this importance for clarifying the application of these works. I recommend the author should provide a more comprehensive overview of these works, some typical works include 10.1109/TAFFC.2018.2874986; 10.1007/s10462-012-9356-9; 10.1007/s13042-017-0705-5; 10.1038/s41467-024-49592-4; 10.1021/acs.nanolett.3c02194, etc.

4) The stability of the model over extended usage is untested. I recommend the authors should provide data on performance degradation over time (e.g., 100+ hours of continuous operation) to validate real-time applicability, if possible.

Author Response

Thank you very much for your valuable feedback; it has been

instrumental in improving our research. Please see the attached pdf file. 
